# Towards a self-sufficient mobile broadband seismological recording system for year-round operation in Antarctica

Alfons Eckstaller[1], Jölund Asseng[1], Erich Lippmann[2] and Steven Franke[1,3]

[1] Alfred Wegener Institute, Helmholtz Centre for Polar and Marine Research, Bremerhaven, Germany

[2] Lippmann Geophysical Instruments (LGM), Schaufling, Germany

[3] Physics of Ice Climate and Earth, Niels Bohr Institute, University of Copenhagen, Copenhagen, Denmark

*Correspondence to*: Steven Franke (steven.franke@awi.de)

**Abstract.** Passive seismic measurements allow the study of the deeper earth beneath the thick Antarctic ice-sheet cover. Due to logistical and weather constraints, only a fraction of the area of the Antarctic ice sheet can be surveyed with long-term or

temporary sensors. A fundamental limitation is the power supply and operation of the instruments during the polar winter. In addition, there is only a limited time window during the field seasons to deploy the stations over the year. Here we present a rapidly and simple deployable self-sufficient mobile seismic station concept. The station consists of different energy supply modules aligned according to the survey needs, measuring duration and survey aim. Parts of the concept are integrated into an already existing pool of mobile stations as well as in the seismological network of the geophysical observatory at Neumayer

III Station. Other concepts and features are still under development. The overall goal is to use these temporary mobile arrays in regions where little is known about local and regional tectonic earthquake activity.



## 1 Introduction

A kilometer-thick ice sheet covers more than 98 % of Antarctica's surface. Therefore, the historical evolution, geological structure and tectonic activity underneath the Antarctic ice sheet are, for large parts, not well known. Continuous, year-round seismic recordings provide a remedy to overcome the direct inaccessibility of the Antarctic continent. The recordings of local, regional and teleseismic earthquakes have been used in various studies. Thus, our present knowledge of the structure of the earth's mantle, lithosphere, and crustal structure underneath the Antarctic ice cover is based on these records. (e.g., Knopoff and Vane, 1978; Nanesi and Morelli, 2001; Ritzwoller et al., 2001; Lawrence et al., 2006; Janik et al., 2014; An et al., 2015 and Lough et al., 2018). However, there is only little seismic activity originating from the Antarctic plate itself due to a low level of tectonic activity (Sykes 1978). Moreover, the low seismic activity is paired with a sparse distribution of seismic instrumentation in Antarctica (particularly in East Antarctica; Figure 1a) and is thus, difficult to verify. Only a few long-term seismic observatories exist. Most of them are constrained to the coastal region and in direct vicinity to research infrastructure (Figure 1).

Many seismic experiments, both using active and passive sources, require numerous seismic stations. The deployments often have to be done in remote and difficult to access areas with very limited power supply and servicing infrastructure. In particular, a long-term AC power supply is not available in most cases. For areas with moderate climatic conditions or when batteries can easily be changed or replaced in regular intervals, this is not a problem. Furthermore, state-of-the-art solar panels provide sufficient electrical power for efficiently charging batteries, enabling instrument operation almost throughout the entire year. In polar regions, however, significant challenges arise in terms of the geographical setting, remoteness and extreme weather conditions, which require a sophisticated power supply design. First, long periods of the dark polar winter with no sunlight available make it necessary to install an additional power supply. If sufficient backup power cannot be realized, it has to be taken into account that data acquisition will stop at some point during polar winter. Second, due to the low temperatures and high discharge of the batteries, it may occur that data acquisition cannot resume when sufficient sunlight is available after the winter break. Additionally, almost all types of batteries show reduced performance at low temperatures and show a substantially reduced effective capacity. Therefore, energy-efficient and for low temperatures adapted renewable systems are required for a sustainable operation in polar environments (Tin et al., 2010).

A major step in this direction was realized within a large international POLENET (Polar Earth Observing Network) within the activities of the International Polar Year (IPY) 2007-2009. In this project, a large number of seismic and GPS instruments were installed in remote sites in Antarctica for several years. The equipment required for POLENET was developed by IRIS (Incorporated Research Institutions for Seismology) PASSCAL (Portable Array Seismic Studies of the Continental Lithosphere) with a focus on a cold-resistant power and communication system that is easy to install and can withstand

Antarctic weather conditions. In this project, large-scale temporary coverage of West Antarctica up to the Transantarctic Mountains, as well as central parts of East Antarctica, was realized for the first time (Figure 1a).

To ensure the continuous extension of seismic coverage in the polar regions, it is essential to find new solutions to the same problems again and again. For polar seismology, it is therefore important to build on previous experience (e.g., from IRIS PASSCAL) to optimize the use of self-sufficient seismometer stations and to find flexible solutions for the different survey areas and deployment lengths. The specifications of the stations must also be realizable with the available resources and be based on long-term scientific goals.


In this article, we describe the concept of an in-house developed mobile and self-sufficient seismological broadband station designed for the extreme demands of the Antarctic ice sheet. A focus lies on (i) the compact modular design and conception of an energy supply to operate under extreme temperatures between -20 to -40° C (on average with slightly warmer temperatures in summer and colder temperatures in winter) and (ii) to present strategies to get the system through the sunless

polar winter. This layout makes the system suitable for long-term operations over several years without regular maintenance and for shorter surveys. Some of the concepts presented here are already in use at numerous seismological stations in western Dronning Maud Land (East Antarctica) operated by the geophysical observatory of Neumayer III Station (Figure 1). Other concepts presented here represent extensions for current and future projects.

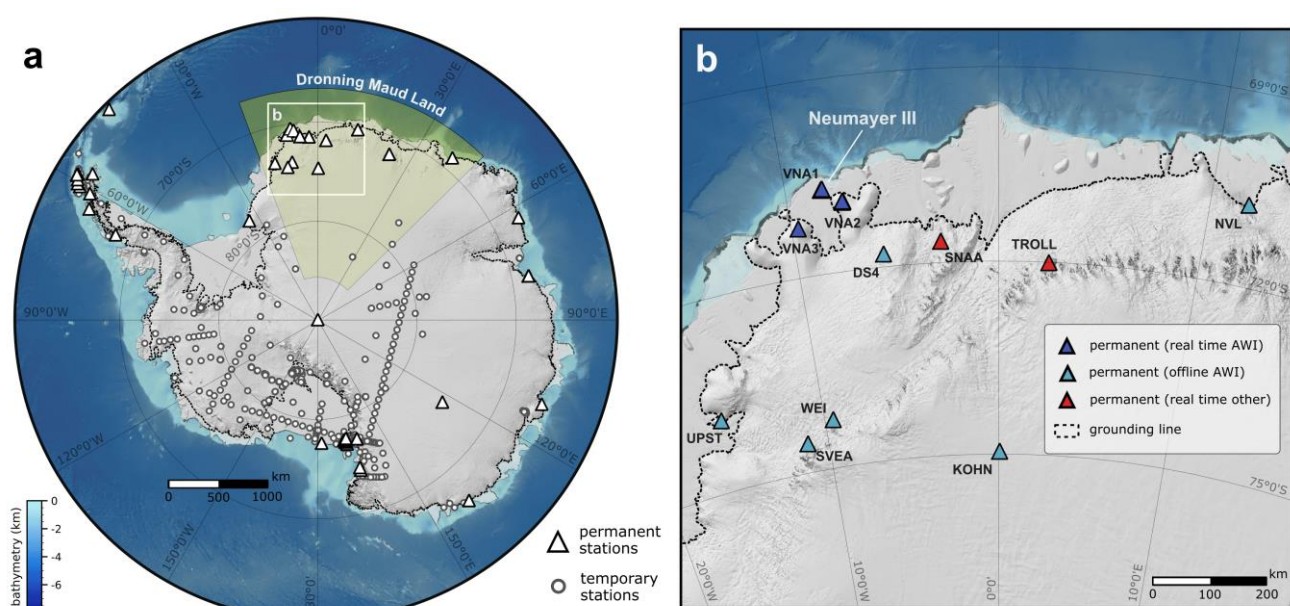


**Figure 1:** (a) Overview of permanent and temporary seismic station distribution in Antarctica (station locations obtained from IRIS GMAP, 2022). Panel (b) shows the distribution of real-time and offline permanent stations operated by the Alfred Wegener Institute (AWI) and

others in western Dronning Maud Land. Note that the majority of temporary stations in (a) were installed by IRIS PASSCAL and that there are potentially more seismic stations in Antarctica whose locations are either not published or where we don't have access to the coordinates.

## 2 AWI's regional seismographic network

The Alfred Wegener Institute, Helmholtz Centre for Polar and Marine Research (AWI), has been operating a local seismographic network for more than two decades in the vicinity of its permanent base Neumayer III Station (Figure 1b). One permanent seismometer is located inside the geophysical observatory close to the station (VNA1), and two other permanent seismometers (VNA2 and VNA3;) are deployed at ice rises at approximately 45 and 85 km distance off Neumayer III Station (Eckstaller et al., 2007). The data quality of VNA2 and VNA3 is substantially better than VNA1 data because the latter is stationed on the ice shelf and the former two are stationed on grounded ice (Figure 1b). Data are continuously transmitted to the base near real-time via high-speed terrestrial data radio. This local network is supplemented by six offline remote semi-permanent seismic stations (Figure 1b; Novolazevskaya NVL, Kohnen Station KOHN, Svea Station SVEA, Forstefjell Nunatak DS4, Weigel Nunatak WEI and Utpostane UPST, respectively). In addition, several temporary single mobile seismic stations or arrays have been deployed in the vicinity of Neumayer III Station for testing purposes and geophysical surveys.

The permanent and mobile temporary seismological stations of the regional AWI seismographic network are located in different glaciological regimes in western Dronning Maud Land and thus are affected by different snow accumulation rates. None of our stations is located in an ablation area. Snow accumulation on the plateau (e.g., KOHN at Kohnen station; Figure 1b) ranges between 15–20 cm per year. By contrast, we observe several meters per year of snow accumulation at the coastal stations (e.g., 3 m per year at VNA3). Depending on the local snow accumulation, the components of the seismological stations, as well as the solar panels or masts, must be relocated to the ice surface, otherwise, they will be buried by the snow over a longer period. This action is mandatory once a year for VNA3 on Sörasen and every 3–5 years for the stations on the plateau. Some stations (e.g., UPST, SVEA, WEI; Figure 1b) are located on nunataks where we observe neither significant snow accumulation nor ablation.

The motivation for developing and optimizing mobile stations is to use them for temporary regional array studies in Dronning Maud Land. The ambition is to use a moving array of seismic stations to acquire data for one to two years and relocate the instruments after that to a new site. Our scientific interests will focus on the analysis of the regional tectonic seismicity associated with a potential neotectonic activity and the analysis of receiver functions to determine the Moho depths and eventually resolve major structural features in the upper mantle in this region.

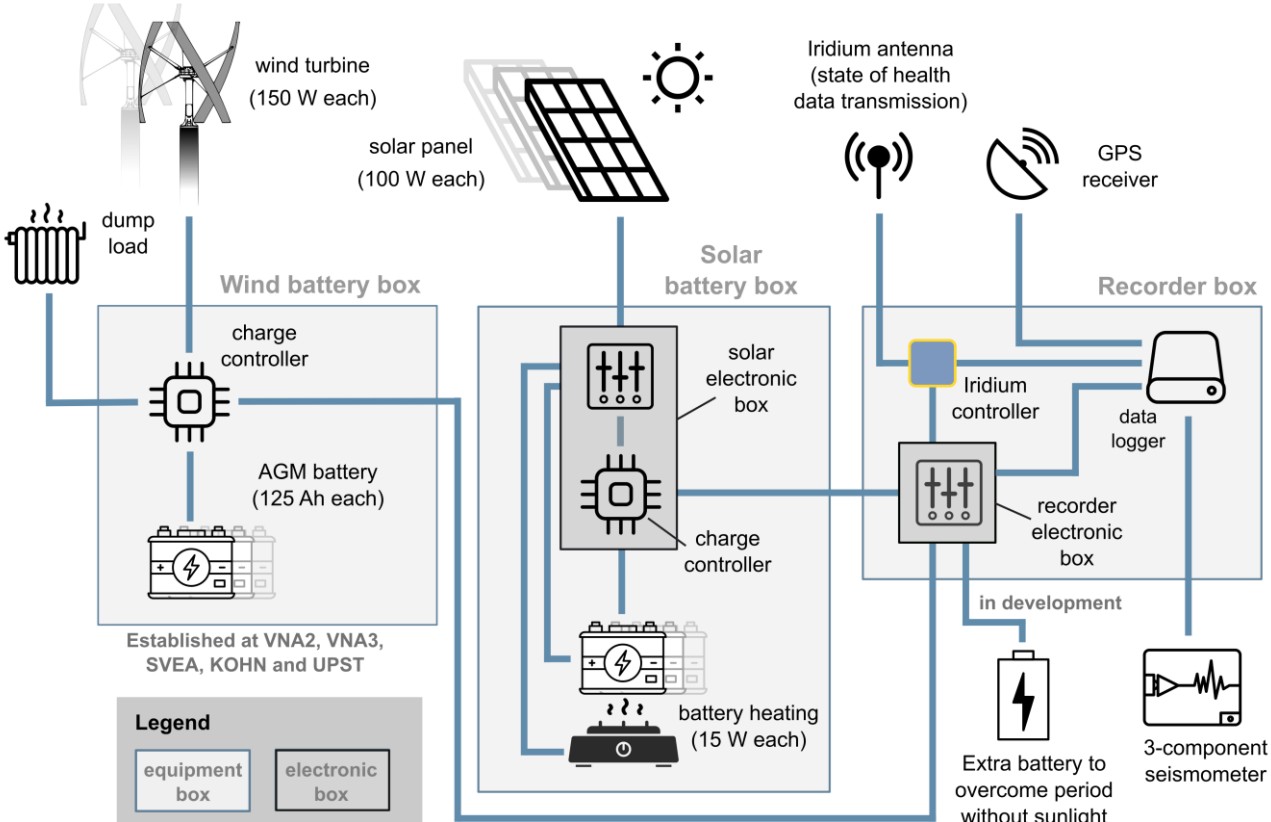

**Figure 2:** Schematic diagram of the instrument layout and power supply concept of our permanent and mobile seismic stations. Note that the station design for our mobile stations does not include the wind battery box. The wind battery box is, however, part of our permanent stations (VNA2, VNA3, SVEA, KOHN and UPST; Figure 1b).

## 3 Concept and instrument design

Our requirements for the mobile seismometer stations must allow a rapid installation and be as modular and compact as possible to enable economic transport and fast deployment and recovery. A single mobile station comprises a solar panel (rack included), a seismometer with casing, one recorder box and one to two battery boxes (Figure 2). For the instrument boxes, we use Peli ISP2 CASES (Inter-Stacking Pattern Cases) boxes because they are waterproof and mechanically stable at low temperatures (Figure 3). The total weight of a single mobile station in its minimum configuration (one seismometer with casing, one seismic recorder, two AGM batteries, one solar charge and iridium controller, two solar panels mounted on one rack, one GPS, one iridium antenna and all cables; Figure 3a–e) is ~ 140 kg and comprises ~ 2–3 m³ storage space.

**Table 1:** Specifications of instruments deployed with AWI's seismic stations. The power drain values represent average values.

| Instrument type | Instrument model | Power drain* | Comment |
|---|---|---|---|
| Seismometer | Guralp CMG-3ESP | 50 mA | |
| | Kinemetrics Metrozet MBB2 | 20 mA | |
| | Streckeisen STS-2 | 46 mA | |
| | Lennartz LE-3D/20s | 50 mA | |
| Seismic recorder | Reftek RT-130 | 83 mA | Favorised for temporary/mobile stations |
| | Quanterra: Q330 + baler or Q330S+ | 50 mA (3-channels) 67 mA (6-channels) | Favorised for permanent stations (Figure 3d), 6-channel configuration only used at VNA2 |
| Iridium controller | XI-202 (XEOS) | 0.1 mA (sleep mode) 50 mA (transmission mode; 2 min./day) | Module for Quanterra (Figure 3d) |
| | SeiDL (SeismicDataLink by SchwaRTech) | same power drain as for the Xeos controller | Custom-made module for RT-130 |
| GPS receiver | GPS 16xHVS (Reftek) | | Power consumption included in the seismic recorder budget |
| Solar charge controller | Blue Sky Solar Boost 3000i | 30 mA (standby) | Not used anymore |
| | Morning Star SunSaver SS-MPPT-15L | 35 mA (standby) | Preferred choice (Figure 4a) |
| Solar cells | Solara S405M36 Ultra 100W | | Mounted on a standard rack (Figure 3a–c, h) |
| | Solara S300M36 Ultra 75W | | Mounted vertically on a mast (Figure 3f) |
| Heating/temperature controller | Minco CT325 Miniature DC | | |
| Wind generator | Twister KD-VK-10 | | Rotor blades are shortened to reduce rotation in strong wind regimes |
| Batteries (lead-acid) | AGM GPL31XT (12V, 125Ah) | | Good capacity/weight ratio (30 kg) |
| Boxes | Peli ISP2 CASES(EU080060-5010, EU080060-4010) | | |

* all electrical consumers are supplied with 12 V

130

We can equip our stations with two data logger types: 3-channels Reftek RT-130 and 6-channels Quanterra Q330S+ (or Quanterra Q330 + baler) recorders (Table 1). Both logger types can be deployed at permanent or temporary mobile seismic stations. The power drain of the recorders is approximately 50 – 83 mA and depends on the number of active channels, sample rate and desired GPS-clock operation. The advantage of the Quanterras is the lower power consumption and the larger storage space. In addition, the Quanterra is easier and more versatile to configure (via a web page GUI from any computer) and has

135 more modern interfaces. However, in contrast to the Reftek data loggers, they are also more expensive.

We commonly use (all three-component) Guralp CMG-3ESP, Kinemetrics Metrozet MBB2 broadband seismometers with a lower corner period of 120 sec and in some cases also Lennartz LE-3D/20s seismometers. The only exception represents UPST station, where we have deployed a Streckeisen STS-2 and a small short period tripartite array. A relevant disadvantage of the Guralp seismometer for the mobile stations is that during transport the mass must be locked to prevent damage. In addition, the instrument must be manually leveled during installation. The advantage of the Metrozet MBB-2 seismometers is the compact design, and the higher transport safety, as it is self-locking and able to center their masses automatically during operation. In addition, the power consumption is very low for an active sensor (20 mA) in comparison to the Guralp seismometer or Lennartz (50 mA).

The solar-powered energy supply system consists of 100 W Solara S405M36 Ultra solar cells and a Morning Star SunSaver SS-MPPT-15L charge controller. Every seismic station is equipped with a state of health (SOH) transmitter that sends the station's operation status in regular intervals once a day via Iridium satellite radio to AWI. For the Quanterra Q330 recorders, we use XEOS XI-202 controllers because they have an existing interface. However, this interface is not available for the newer Q330S+ recorders. For the RT-130 we use a custom-made iridium controller (SeiDL - Seismic Data Link) to influence all parameters and configurations. For example, it gives us the possibility to transmit data from additional environmental sensors, such as wind, temperature, solar radiation, current and voltage (if available). This controller was developed by Arne Schwab (SchwaRTech, based near Bremen, Germany), and also uses the Iridium short burst data (SBD) transmission technique. The average power drain of our iridium controllers is very low with 0.1 mA in sleep mode and 50 mA in transmission mode (2 minutes per day). The overall power consumption of an entire single mobile station is 4 Ah per day and 1447 Ah per year. The calculation is based on a station design that comprises a Guralp CMG-3ESP seismometer and a Reftek-130 data logger. For the wiring of all devices, we have moved away from PVC insulated cables, as they are too brittle at low temperatures. Now we use almost exclusively more flexible cables with PE or PU insulation with improved UV and cold resistance. A complete list of the specifications of the instruments is provided in Table 1.

### 3.1 The battery box

### 3.1.1 Battery box configuration

Each battery box consists of two 125 Ah AGM (Absorbent Glass Mat) lead-acid batteries, a charging controller, and additional control electronics (Figure 3e). The advantage of AGM batteries is their good performance at low temperatures and that they are not categorized as dangerous goods for transport. The batteries are placed on an aluminum plate with two 10 or 20 W Silicone heating foils attached to its bottom side. The heating foils are underlaid with a thin heat-resistant layer to prevent the eventual melting of the insulation foam. The box is connected with one or multiple 100 W solar panels as input power. The

solar panels can be mounted on standardized aluminum racks for deployment on snow and solid rock, which the AWI workshop

170 had manufactured (Figure 3a). The racks are mechanically robust and can resist high wind speeds (25–50 m/s) despite their lightweight if tied to anchors buried in the snow or stone bolts. For thermal isolation, we use Alveobloc panels (produced by Sekisui Alveo) with a thickness of 8–10 cm as thermal insulators for the box interior. The isolation panels are available in different densities. We use a denser and harder type (Type 1700; 60 kg/m$^3$) for the bottom layer of the box to bear the heavy weight of the batteries without deformation. For the remaining isolation, we use a lighter and softer Alevobloc type (Type

175 3600; 28 kg/m$^3$). The material can easily be cut with a table saw into exactly fitting blocks. All electronic components, the charging controller, heating electrics and additional control electronics are installed inside a compact box (here referred to as the solar electronic box; Figure 2) that just fits beside both accus (Figure 3e and Figure 4a,c). All necessary electronic units are installed on DIN rails, which allows compact and structured cabling.

180

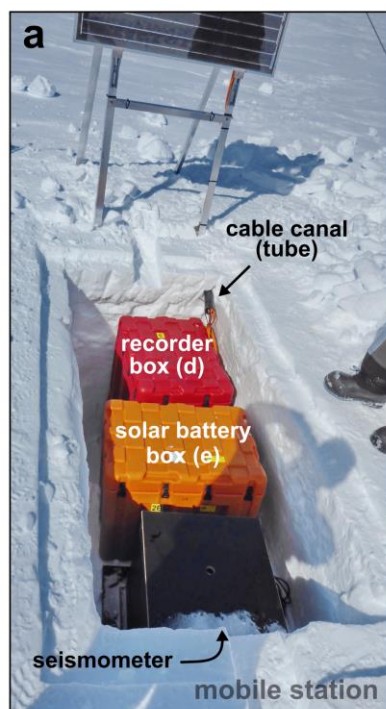

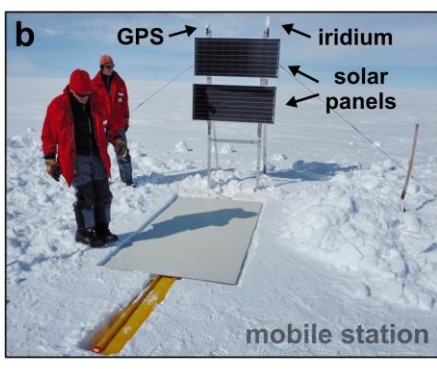

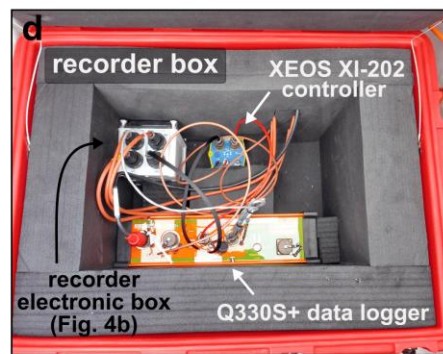

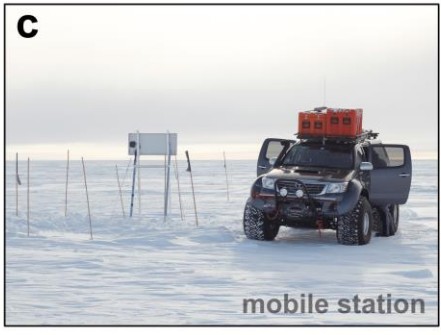

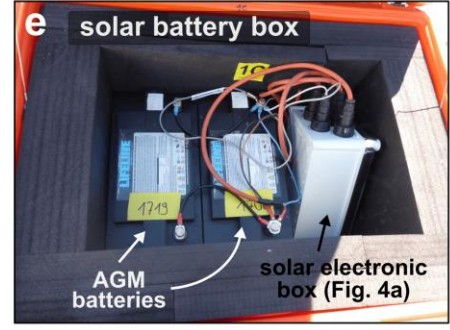

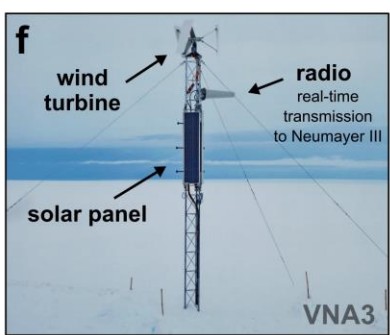

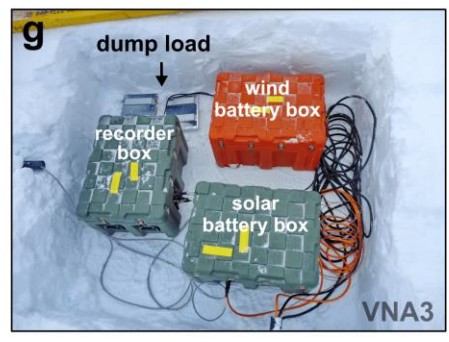

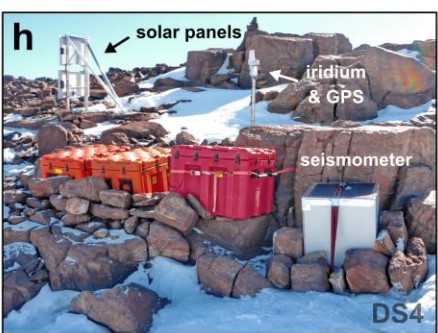

**Figure 3:** Photographs of the station design from several deployments in western Dronning Maud Land (East Antarctica). Panels (a–c) show the setup of the mobile stations on the ice surface. Panels (d) and (e) show the recorder and solar battery boxes from the inside. The permanent real-time station setup of VNA3 is shown in (f) for the ice surface and in (g) for the ice cave. Panel (h) shows the station layout of DS4, which is deployed on a rock base. Photo credits: (a,b, d-h) Jölund Asseng; (c) Steven Franke.

### 3.1.2 Self-discharge protection

The key feature in the solar electronic box in our battery boxes is a special control electronics (solar controller; Figure 4a,c) to guarantee that battery charging will resume after the several week-long break during polar winter. Due to the inevitable self-discharge (if not connected to a power source), the battery voltage can drop below a critical value of approximately 8–9 V. A voltage level below this threshold implies the risk that the charging controller cannot resume operation again. Without the

power supply from batteries, charging controllers cannot operate with the solar panels' output. Additionally, even in standby mode when the LVD (low voltage disconnect) control disconnects the recorder and seismometer, the controller continuously drains current from the batteries which can cause additional voltage decrease. Therefore, our control electronics will disconnect the charge controller from the batteries when dropping below a critical value, which we set to 11.0 V. At the same time, the solar panels are directly connected to the batteries (Figure 2). This enables the batteries to be charged directly with the electric power of the first sunlight after the winter break. When the battery voltage rises above the threshold of 13.0 V, the charge controller is reconnected to the batteries. At the same time, the solar panels are reconnected to the charge controller. This principle enables a safe return to the normal operation mode. The solar control and its electronics were designed and manufactured by Erich Lippmann (LGM) and inserted in a small green enclosure inside the solar electronics box (Figure 4a,c).

### 3.1.3 Battery heating

All kinds of batteries show better performance and higher capacity if they are not exposed to too cold temperatures. Moreover, the electrolyte freezing point is significantly lower for fully or partly discharged batteries (Figure 5). Nonetheless, if the AGM batteries we use freeze, they are not destroyed, but simply stop providing energy and start working again at higher temperatures. Therefore, we realized the option for battery heating if sufficient power is available (excess energy from the solar panels). Battery heating will only be enabled if the battery voltage has exceeded an upper threshold value of 14.5 V and will be disabled if falling below the lower threshold value of 13.0 V. This is accomplished by using a voltage guard relay from MRS Electronic. The threshold voltages can be freely programmed and set to desired values. The module is designed for automotive applications and is thus very robust and reliable. For heating control, we use the Minco CT325 Miniature DC Temperature Controller (Figure 4a,c), which permits a heating current of up to 6A. This implies that the heating plates have to be configured in a way that not more than 6A can flow, which corresponds to a maximum heating power of 66 W at 12V. Therefore, our heating plate set includes 2 times 15 W. This is sufficient for the heating plates under the bottom of the aluminum plate where the batteries are placed on. The sensor is a PT-100 element that is mounted on an aluminum bar that is attached to the aluminum plate for good heat conduction. We set the desired battery temperature to +20° C. This temperature may only be reached during summer, but it will then keep the batteries during this period relatively warm.

### 3.2 The recorder box

The recorder box is of the same type as the battery box, with a slightly lower height and is thermally insulated in the same way (Figure 3d). Besides the data recorder and the SOH control modem, there is a recorder electronic box containing a backup battery management controller (BBat-controller) for battery box management (Figure 4b,d). Two solar rechargeable AGM batteries and one backup battery can be connected. For two connected solar battery boxes, the BBat-controller acts as two ideal Schottky diodes which are switched in a way that power comes only from the battery box with the higher voltage. In case both voltage levels are equal, the two battery boxes provide the same amount of current. If both solar battery boxes are disconnected by their internal LVD or both voltages drop below approx. 8.9 V, the power supply will be almost simultaneously switched to

the backup batteries. It will switch back to solar batteries if one of the voltages rises back again above 10.7 V. This MOS-Fet-based switching electronics were designed and manufactured by Erich Lippmann and utilizes the LTC4416 controller chip which is widely used for backup power supply systems. Furthermore, our circuit design prevents the flow of current from one battery box to the other. The current drain is max. 0.3 mA, thus extremely low and does not play any significant role in the total power consumption considerations. With some additional minor modifications, the total battery capacity of the system can be extended by additional battery boxes.

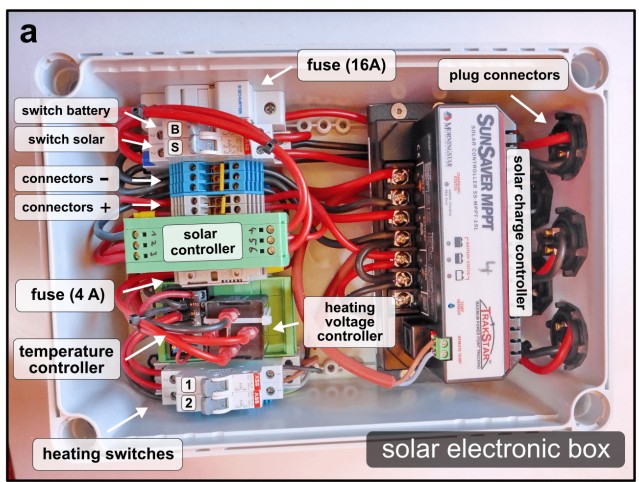
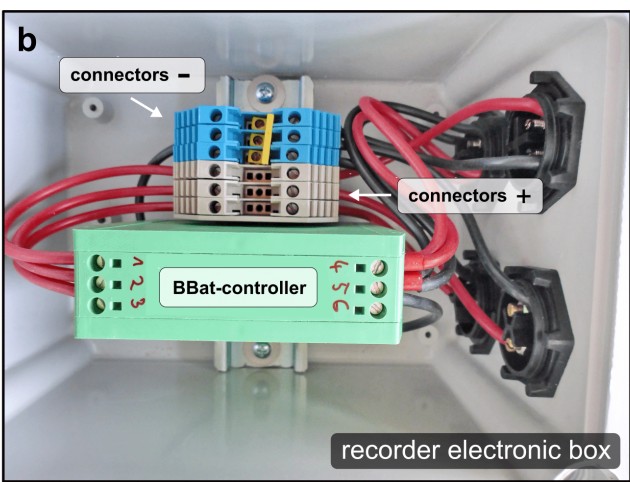
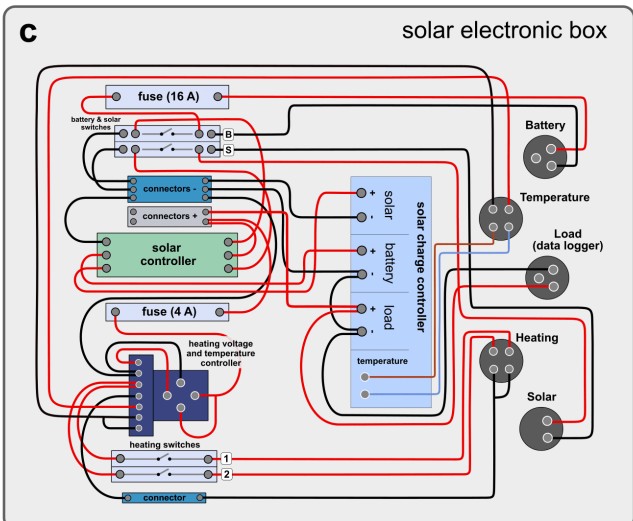
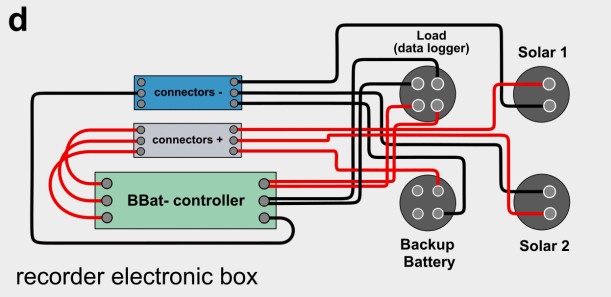

**Figure 4:** Electronic boxes indicated in Figure 2 and Figure 3d,e. Panel (a) shows the solar electronic box containing the solar charge controller, the solar controller and additional electronics. Panel (b) shows the recorder electronic box and contains the backup battery management controller (BBat controller). The schematic wiring of the components for the solar and recorder electronic box is shown in panels (c) and (d) respectively.

All necessary connecting cables are connected with the connectors on the rear side of the recorder box. We decided to use the so-called *Reftek standard* for sensor input, which means that the pin configurations of the sensor connectors correspond to Reftek specifications. If using a Quanterra Q330, the cable from the rear sensor connectors needs to be configured appropriately for this recorder. This will allow the connection of both of our sensors without needing an extra connector or adapters for the specific recorder, which allows higher flexibility, and reduces the deployment time and susceptibility to errors.

## 4 Discussion

### 4.1 Overcoming the polar winter gap

The absence of the sun in the polar winter creates a supply gap of input energy, which usually leads to a data acquisition gap if solar cells are the only energy source and the battery capacity is not high enough to provide energy for several weeks. We have developed a concept for our mobile stations to restart the data recording after the polar winter reliably. Beyond, there are different ways to bridge this period, and the advantages and disadvantages of these systems are discussed below.

#### 4.1.1 Backup batteries

Based on our experience, when considering a usable effective capacity of 30% at -40°C (Figure 5), one 125 Ah AGM battery can provide power for approximately 14 days at polar winter onset (if solar panels are the only power source). Hence, using two AGM batteries will not ensure recording for more than 4 weeks without recharging. This implies an inevitably long recording break during polar winter unless high-capacity backup batteries are added. These could either be high-capacity battery types, such as Lithium Thionyl Chloride (LTC) primary cells or rechargeable LiFePO4 accumulators. The concept of using Li-based batteries has already been successfully demonstrated in Antarctic campaigns with the PASSCAL instruments (e.g., Hansen et al., 2015). Further reasons why we prefer a Li-based battery with a high energy density to the use of additional AGM batteries are the following. First, the number of AGM batteries needed to last through the polar winter would be very high and add a lot of transport weight. Theoretically, of course, these could be recharged over the summer. The problem, however, is that at the end of the polar night, there is little light and thus little current flow available to charge a large total capacity. This can result in the entire system being very slow to get above the minimum voltage to start data acquisition, thus extending the data gap period.

The power bridging concept during the polar winter gap with backup batteries has been so-far only applied in a proof-of-concept testing period with AGM batteries. The usage of Li-based batteries has not been implemented so far. The main reason for this is that the transport and storage of these batteries are restricted as they have to be treated as dangerous goods. Especially transport by aircraft may sometimes become almost impossible. The second reason is that they are still very expensive unless they are produced in higher quantities or if further developments make them more affordable. However, in principle, all our

mobile stations could also be equipped with Lithium based backup batteries. Since the lithium batteries cannot be recharged, this solution is suitable for temporary applications designed for 1–2 years.

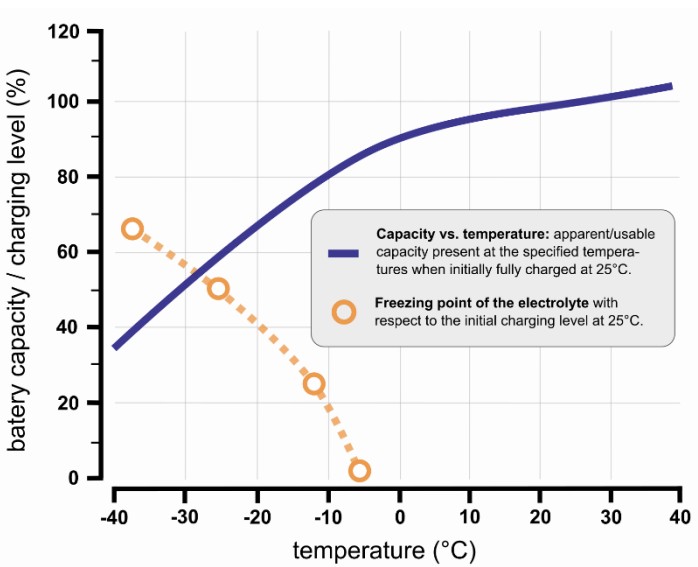

**Figure 5:** AGM battery performance under cold conditions. The dark blue line represents the effective usable battery capacity at the indicated temperature, based on a fully (100%) charged battery at 25°C. The curve applies to an approximate electrical current flow corresponding to 5% of the total capacity of the battery (Lifeline Technical Manual, 2019). The orange circles represent the freezing point of the electrolyte. The freezing point depends on the battery charging level at 25°C. Thus, a battery charged to 50% of its total capacity at 25°C will freeze at -25°C and at -13°C if charged to only 25% (Lifeline Technical Manual, 2019).

### 4.1.2 Additional wind generator

Wind generators are an alternative energy source that is independent of the light conditions in the polar winter. This option has been implemented for other non-permanent seismic stations (e.g., Anandakrishnan et al., 2000; Contrafatto, et al., 2018) as well as for long-term seismic stations in our network (VNA2, VNA3, SVEA, KOHN, UPST; Figure 1b and 3f). Here, we use helical horizontal axis wind turbines (HAWT) which consist of three rotor blades (for further information and a schematic illustration of helical HAWT wind turbines see Peng et al., 2021). We modified the smallest available version (which generates 300 W power) with wings half the length to reduce the mechanical stress on the system. This reduced generator version produces ~ 150 W power. One advantage of this kind of wind generator is that the bearings do not require regular oiling. The wind generator at VNA3 ran for instance for 5 years without maintenance. However, we still see potential in the control of the generator when exposed to very strong winds, especially when the batteries are fully charged. The general principle is that

when the batteries are fully charged and the excess wind energy is dissipated through resistors (dump load; Figure 2 and 3g), the rotation is reduced simultaneously. Principally, this option can be integrated into our mobile stations and could enable data recording over the entire year. However, this concept has several disadvantages, especially for the recording of seismological data. The vibrations caused by the wind generators are transmitted to the ground or snow and thus recorded by the seismometer. Depending on the coupling between the wind generator and the ground and the distance between the wind generator and the seismometer, the seismological data may be disturbed or even unusable. In addition, wind generators are mechanically very susceptible to these extreme conditions and the strong Antarctic winds. However, they are currently indispensable for a long-term energy supply over many years. In addition, the time required to set up a seismological station increases significantly with the installation of a wind generator, considering the short installation time of the solar panels, seismometer and instrument boxes. Wind turbines can, in principle, also be used for effective battery heating during the polar night. This requires, however, additional equipment (and thus cargo) and represents another source of system failure.

### 4.2 Choosing the appropriate solar charge controller

In the development stage of the mobile stations, we used two different solar charging controllers, a Blue Sky Solar Boost 3000i and a Morning Star SunSaver SS-MPPT-15L. Both controllers are maximum power point trackers (MPPT), which show high efficiency and can produce sufficient charging current even in weak or diffuse daylight conditions. The Blue Sky SB 3000i offers a variety of features and needs comprehensive programming for the setup. It can display battery voltage and charging current and the maximum charging voltage among other features. The specific temperature coefficient for AGM batteries can be programmed, and the LVD can be arbitrarily chosen. However, the variety of available settings and vulnerability to incorrect programming leading to total system failure is a problem if the mobile stations are not installed by trained personnel. Moreover, we experienced during our testing period, that some Blue Sky controllers lost their programming when battery voltage was very low during winter. Therefore, we have chosen not to use *Blue Sky* charge controllers for our mobile stations to minimize the susceptibility to errors during programming, installation or in the polar winter to enable also less-trained staff to deploy the stations. For all mobile seismic stations, we, therefore, use the simpler Morning Star Sunsaver charge controllers. They offer two LVD voltages to choose from for recorder shut down at low voltage conditions. Until now, we had an excellent experience with these controller types, which have been proven to work at very low temperatures between -20 to -40°C properly.

### 4.3 The electrostatic discharge problem

If the mobile stations are placed on the ice and not on the few outcropping rocks, the system is vulnerable to static charging. Since snow, firn and ice are very poor electrical conductors, there is almost no possibility to find a suitable ground to prevent electrostatic discharge of high currents. The electric charge itself is caused by all station elements positioned outside the snowpit (solar cells and their racks, GPS and Iridium). For the GPS, it is possible to operate it under a limited snow cover thickness. However, the solar cells and the iridium must necessarily be installed on the surface. In our testing period, we

attempted to create a mass for the electrical compensation with large metal elements, which we have buried in the snow, but only with moderate success. Additionally, by connecting all instruments and equipment to the same potential (minus), the damage or failure rate of the system due to electrical discharge was reduced. It should be noted that this is not possible with all solar charge controllers (for example, it was not possible with the Blue Sky solar charge controller but possible with the Morningstar solar charge controller). However, still, the problem currently generally remains and can cause long-term damage to electrical equipment or, in rare cases, system failure.

## 4.4 Comparison to and lessons learned from other seismic surveys

The development of self-sufficient seismic stations has been strongly promoted for the extreme conditions of the polar regions in the last two decades. The component design and the available resources of various temporary or long-term year-round seismic measurements in Greenland (e.g., Dahl-Jensen et al., 2010) and Antarctica (e.g., Hansen et al., 2015) differ between surveys. However, some of the concepts have gained acceptance and many useful recommendations for future campaigns and networks have emerged from numerous scientific publications and field reports over time. Our concept of a fast to deploy, compact, modular self-sufficient mobile seismic station aims to use the limited time in the field efficiently and is based on many of the experiences described in the literature, which we discuss in the following.

Since 2009, the GreenLand Ice Sheet monitoring Network (GLISN), has been initiated to monitor all types of earthquakes with broadband seismometer stations in Greenland (Dahl-Jensen et al., 2010). Four of the 33 stations are deployed on the ice-sheet interior (Veitch and Nettles, 2012). Here, the power system of one of the stations consists of a large number of batteries and solar cells (26 6V AGM batteries and nine 80W 12V solar cells) to ensure long-term operation (Toyokuni et al., 2014). This configuration enables a year-round operation but requires a large amount of heavy equipment. Moreover, a large portion of the batteries are not required for summer operation but consume high logistical capacities. A smaller number of batteries in combination with solar cells, wind generators (and a dump load for excess energy if the batteries are fully charged) as well as a low-voltage disconnector to protect the batteries from deep discharge has been used in a survey with six broadband seismic stations in West Antarctica in 1998 (Anandakrishnan et al., 2000). Although during the first year of deployment the total time of data recording was only 50 %, some of the stations were able to operate throughout the year. The authors suggest that longer uptimes can be achieved by improving the insolation of the battery boxes, which is a concept that we have implemented in our system. A similar approach in system design is introduced by Contrafatto et al. (2018). A major advancement in continuous seismic recordings in Antarctica came with the deployment of the 30-station Gamburtsev Antarctic Mountains Seismic Experiment (GAMSEIS) array on the East Antarctic plateau (Hansen et al., 2015). The novel station design was developed by IRIS-PASSCAL for polar applications (Johns et al., 2006) and enabled the deployed stations to operate year-round with the usage of lithium backup batteries in the winter. IRIS' successful continuous development strategies of the winter data collecting

capability have increased the data recovery rate from < 50 % to more than 90 % within five years. The setup was used by Heeszel et al. (2013) and enabled a total data recovery rate of 93 %.

## 5 Future visions

In addition to the concepts currently under development, we also have ideas for subsequent developments. Above all, we see much potential in optimizing battery management and input energy management. For example, a multiple-battery option would

be desirable, in which individual batteries are charged step by step after the polar winter when the current flow is low so that a high voltage is available quickly. It would also be desirable to disconnect deeply discharged batteries from the overall system. In terms of input energy management, a variant is conceivable in which a wind generator is switched on exclusively in the polar winter. This would close the energy gap in the polar winter (with the acceptance of increased noise in the data) and generate no noise during the summer season while recording data. Another possibility to reduce the noise influence and

material stress of the wind generators would be to switch on the wind generator only for a particular time when the total voltage of the batteries drops below a certain range.

## 6 Summary and conclusions

We have presented a fast and easy to deploy modular, compact, mobile and self-sufficient seismometer station concept for the polar regions. Due to its modular design, it can be used in various ways, for example, for short-term deployment as an array

over 1–2 years or as a long-lasting permanent station. The energy supply can be adapted as required using the modular cascading of battery boxes, wind generators, solar cells or backup batteries, which enables optimum use of limited resources. The stations' modules are designed so that only the cables have to be connected in the field. Parts of the concepts presented here are already in use as part of the extended seismology network of the Neumayer III Station. Our system concept is not specifically limited to the application to seismology stations (except for noise suppression) and can also be extended by

additional instruments with low power consumption (e.g., to monitor environmental parameters). Moreover, it is a suitable system for managing the power supply for all types of self-sufficient measuring systems in polar regions.

## Acknowledgements

We especially thank Ulrike Windhoevel for her editorial contribution to the manuscript. Moreover, we would like to thank the scientific workshop of the Alfred Wegener Institute for their support in the production of custom-made components, such as

cable connectors, racks for solar cells and wind generators and the supply of special tools for the field. We would like to note that many of the concepts presented here are inspired by the long-standing and intensive efforts of IRIS (Incorporated Research Institutions for Seismology) PASSCAL's (Portable Array Seismic Studies of the Continental Lithosphere) engagement in the

polar regions (https://www.passcal.nmt.edu/content/polar). The icons used in Figure 2 were downloaded from the *Noun Project* (https://thenounproject.com/) and we acknowledge the following creators: solar cell by *monkik*; heater by *lastpark*, extra

battery by *Irman*, AGM battery by *Rusmaniah*, seismometer by *faisalovers*, stove by *hasanudin*, gps receiver by *Arthur Shlain*, data logger by *iconsmind.com*, antenna by *iconpixel*, control unit by *Delta*. For more information about the geophysical observatory at Neumayer III Station, please refer to: https://www.awi.de/en/science/geosciences/geophysics/research-focus/observatories-long-termeasurements.html.

## Author Contributions

AE, JA and EL led the development of the energy management concept and mainly manufactured the mobile seismic stations. AE, JA and SF wrote the manuscript.

## Financial Support

Steven Franke was funded by the AWI Strategy fund and the German Academic Exchange Service (DAAD) programm "Forschungsstipendien für promovierte Nachwuchswissenschaftlerinnen und -wissenschaftler (Kurzstipendien)".

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
