# Peer review of "Towards a self-sufficient mobile broadband seismological recording system for year-round operation in Antarctica"

_Geoscientific Instrumentation, Methods and Data Systems, 2022_

## Author Comment (AC1)

First of all, we would like to thank Alex Brisbourne for his time and constructive, thorough and helpful suggestions, which are each addressed below. Our responses are organized in the following color code:

- the original text of the reviewer (black)
- response to the reviewer comments (blue)
- text removed from the main article (lila)
- text added to the main article (green)
* * *
**Review of "A self-sufficient mobile broadband seismological recording system for year-round operation in Antarctica" by A. Eckstaller et al., 2022, Geoscientific Instrumentation**

**Alex Brisbourne, April 2022**

Eckstaller et al. present an overview of their mobile seismic station system for use in the polar regions. The group have developed a relatively lightweight and mobile system for use in a modular manner with different seismic and recording systems. The manuscript covers concepts, requirements and solutions.

The paper is well written and easy to follow. It provides a lot of useful ideas and concepts which practitioners embarking on similar deployments will find useful to be highlighted prior to starting such projects and therefore forms a useful piece of work.

**Major comments**

The system has a lot of similarities with the IRIS Passcal system used for the Polenet Project which has become the standard for many deployments in Antarctica. However, it takes until the acknowledgements for this to be recognised. In addition, there is little mention in the introduction of Polenet or IRIS. For example, all the white circles on Fig. 1a are Polenet stations and the authors do a disservice to the work of these groups to have achieved the station coverage of West Antarctica that Figure 1 highlights. I would suggest a paragraph in the introduction covering: The IRIS Passcal system; the relationship between this system and the IRIS Passcal system and why the concept presented here I needed; the achievements of the Polenet project. Again, in section 4, I suspect that IRIS Passcal now have some impressive year-round data recovery rates. I would like to see these mentioned for comparison and perhaps some comments on how this work advances that of Passcal.

We gladly accept this suggestion and we agree with the reviewer's opinion that the developments of IRIS Passcal and the achievements of the POLENET project should be listed much earlier.

The achievements of the POLENET project and a statement why this and other systems are still needed were added the following paragraphs to the introduction:
"A major step in this direction was realized within a large international POLENET (Polar Earth Observing Network) within the activities of the International Polar Year (IPY) 2007-2009. In this project, a large number of seismic and GPS instruments were installed in remote sites in Antarctica over several years. The equipment required for POLENET was developed by IRIS (Incorporated Research Institutions for Seismology) PASSCAL (Portable Array Seismic Studies of the Continental Lithosphere) with a focus on a cold-resistant power and communication system that is easy to install and can withstand Antarctic weather conditions. In this project, large-scale temporary coverage of West Antarctica up to the Transantarctic Mountains, as well as central parts of East Antarctica, was

realized for the first time (Figure 1a).

To ensure the continuous extension of seismic coverage in the polar regions, it is essential to find new solutions to the same problems again and again. For polar seismology, it is therefore important to build on previous experience (e.g., from IRIS PASSCAL) in order to optimize the use of self-sufficient seismometer stations and to find flexible solutions for the different areas and deployment lengths. The specifications of the stations must also be realizable with the available resources and be based on long-term scientific goals."

Furthermore, we have now added a statement at the end of Section 4.4 regarding the increasing data recovery rates from IRIS Passcal:
IRIS' successful continuous development strategies of the winter data collecting capability have increased the data recovery rate from < 50 % to more than 90 % within five years. The setup reported by Heeszel et al. (2013) enabled a total data recovery of 93 %.
* * *
I feel that there are details missing that would make this a much more useful paper. A number of the statements are subjective. I would like to see a table of specifications for components listing power draw and weight for example and a way of understanding the relative power-cost of individual components (sensor and data logger are reported), such as the XEOS or Solar controller. How about example overall station weight/volume etc.

We agree with the reviewer, that information on power draw and weight is missing and very useful for the reader. We have now integrated the power consumption of all electrical consumers (including iridium and solar charge controllers)l in Table 1. The power consumption is reported in ampere (A) with an input voltage of 12 V.

Furthermore, we have made the following estimates on the weight of our individual components:

Peli box: 10kg
AGM battery: 30kg
Seismometer: 5kg
Seismometer casing: 10kg
Data recorder: 5kg
Electronics: 5kg
Kabel + protection: 10 kg
Solar cell: 5kg
Solar cell rack: 15kg

For a single mobile station in the smallest possible configuration, we use two Peli boxes, two batteries, one seismometer (including casing), one data recorder, one solar cell rack and two solar cells. This equals 140 kg and comprises ~ 2-3 $m^3$ storage space.

We added the following sentence to the first paragraph of Section 3:
"The total weight of a single mobile station in its minimum configuration (one seismometer with casing, one seismic recorder, two AGM batteries, one solar charge and iridium controller, two solar panels mounted on one rack, one GPS, one iridium antenna and all cables; Figure 3a-e) is ~ 140 kg and comprises ~ 2 m3 storage space."
* * *
The title including "year-round" may be a stretch as it seems that this concept has yet to be fully established for year-round recording. Maybe that's why the word operation is used? Perhaps a little disingenuous.

Thank you very much for this remark, and we agree. We decided to change the title to "**Towards a self-sufficient mobile broadband seismological recording system for year-round operation in Antarctica**" to be in line with the comment from another reviewer and highlight that we are on the way but not there yet to record data year-round.
* * *
In many ways the manuscript leaves a lot of questions unanswered, mostly being why did the developers use this controller or that modem? Were other systems tested and ruled out? You could save future practitioners some effort by stating why these units were used over others (not necessarily having to publish manufacturer's names).

This is a very good hint, which we gratefully accept. We have added some additional information comparing the instruments (seismometer, recorder, Iridium controller, etc.) and evaluating their advantages and disadvantages in Section 3.

Added text is marked green and existing text black:
"We can equip our stations with two data logger types: 3-channels Reftek RT-130 and 6-channels Quanterra Q330S+ (or Quanterra Q330 + baler) recorders (Table 1). Both logger types can be deployed at permanent or temporary mobile seismic stations. The power drain of the recorders is approximately 50 – 83 mA and depends on the number of active channels, sample rate and desired GPS-clock operation. The advantage of the Quanterras is the lower power consumption and the larger storage space. In addition, the Quanterra is easier and more versatile to configure (via a web page GUI from any computer) and has more modern interfaces. However, in contrast to the Reftek data loggers, they are also more expensive.

We commonly use (all three-component) Guralp CMG-3ESP, Kinemetrics Metrozet MBB2 broadband seismometers with a lower corner period of 120 sec and in some cases also Lennartz LE-3D/20s seismometers. The only exception represents UPST station, where we have deployed a Streckeisen STS-2 and a small short period tripartite array. A relevant disadvantage of the Guralp seismometer for the mobile stations is that during transport the mass must be locked to prevent damage. In addition, the instrument must be manually leveled during installation. The advantage of the Metrozet MBB-2 seismometers is the compact design, and the higher transport safety, as it is self-locking and can center the mass internally. In addition, the power consumption is very low for an active sensor (20 mA) in comparison to the Guralp seismometer or Lennartz (50 mA).

The solar-powered energy supply system consists of 100 W *Solara S405M36 Ultra* solar cells and a *Morning Star SunSaver SS-MPPT-15L* charge controller. Every seismic station is equipped with a state of health (SOH) transmitter that sends the station's operation status in regular intervals via Iridium satellite radio to AWI. For the Quanterra Q330 recorders, we use XEOS XI-202 controllers, because they have an existing interface. However, this interface is not available for the newer Q330S+ recorders. For the RT-130 we use a custom-made iridium controller (SeiDL - Seismic Data Link) to have an influence on all parameters and configurations. For example, it gives us the possibility to transmit data from additional environmental sensors, such as wind, temperature, solar radiation, current and voltage (if available). This controller was developed by Arne Schwab (SchwaRTech, based near Bremen, Germany), and also uses the Iridium short burst data (SBD) transmission technique. For the wiring of all devices, we have moved away from PVC insulated cables, as they are too brittle at low temperatures. Now we use almost exclusively more flexible cables with PE or PU insulation with improved UV and cold resistance. A complete list of the specifications of the instruments is provided in Table 1 "

**Minor comments**

2 and elsewhere – 3k seismometer – do you mean 3-Component?

We agree with the reviewer that "3-component seismometer" is the correct term and believe the reviewer is referring to Figure 2. This has been changed throughout the document.

"3-k" to "3-component" (in Figure 2 and also in the text)
* * *
Table 1 – I am not sure I would call the table or column 2 "Instrument specifications", it is more the model numbers. I do however feel that at some stage (probably in the appendix) more detailed specifications would be welcome, such as temperature rating, power draw etc).

Changed "Instrument specification" to "Instrument model".
* * *
L108 – What is the power drain of the Morningstar 15L?

The power drain of the Morningstar solar controller is 35 mA per day. This has now been included in Table 1.
* * *
L110 – Likewise, what is the power requirement of the XEOS over a season?

The average power drain for the XEOS and SeiDL iridium controller is 0.17 mA, which corresponds to 4 mAh per day and 1.49 Ah per year. This is the sum of the power consumption in sleep mode and transmission mode for one (2 minutes) transmission per day.
* * *
L122 – can you quantify "high winds"?

We define "high winds" as wind speeds beyond the range where it is feasible to work outside (25-50 m/s). We added this number now in the text.
* * *
L159 – not sure what "into the drilling hole" means

We mean that the sensor is placed in a hole of an aluminium bar.

"The sensor is a PT-100 element that is placed in a hole of an aluminum bar that is attached to the aluminum plate."
* * *
L173 – good to quote power drain in Ah or Ah/day

We agree with the reviewer that quoting the power drain in Ah makes sense to provide an overview of the power consumption per day or year. However, at this particular place in the text, we believe that it doesn't make a lot of sense. The switching electronics mentioned in line 173 are operating very rarely and can therefore be neglected in our energy budget. Nonetheless, the reviewer makes a good point and we are referring to the power drain in Ah in another place in our responses to the reviewer's comments.

We also want to note that in that line we made a mistake in the original version, where 0.3 A is incorrect and should be 0.3 mA.
* * *
L174 – it would be good to know how the overall power budget is distributed amongst the components.

We agree and we have added an extra column in Table 1 with the power drain of all electrical consumers. These specifications refer throughout to operation with 12 V. The consumption in watts can therefore be calculated independently. For a power drain over an entire season we choose the following values:

**Power drain (A)**
Seismometer: 50 mA
Data logger: 80 mA
Iridium controller 0.17 mA (average value including a 2 minute transmission time per day at 50 mA)
Solar charge controller: 35 mA

**Power drain over time (Ah)**
The total power drain of all electric consumers is 165 mA, which would correspond to 4 Ah per day and 1447 Ah per year.

We added the following text at the end of section 3:
"The overall power consumption of a single mobile station is 4 Ah per day and 1447 Ah per year. The calculation is based on a station design which comprises a Guralp CMG-3ESP seismometer and a Reftek-130 data logger."
* * *
L194 – Again, IRIS Passcal have addressed and solved this issue in one way but it isn't mentioned.

We believe that by "this issue" the reviewer is referring to the gap in data logging in winter when operating with a few AGM batteries and solar cells. The reviewer is right and IRIS PASSCAL has already successfully bridged this gap with Li-based batteries. However, we would like to point out that we have already mentioned this in section 4.4 (L272-274 in the initial submission):

"The novel station design was developed by IRIS-PASSCAL for polar applications (Johns et al., 2006) and enabled the deployed stations to operate year-round with the usage of lithium backup batteries in the winter."
* * *
L227 – another issue that could be highlighted is that wind strengths are highly variable spatially and in my experience the manufacturing tolerance of wind generators tends to be poorly managed so two adjacent supposedly identical units can respond differently to wind strength. Do the authors have any experience of this that could be included?

We agree with the reviewer that this could be the reason why apparently identical wind generators behave differently under the same conditions. At VNA2, we actually use two identical wind generators, where we have had a couple of break downs of one of them, whereas the neighbouring wind generator worked fine. Unfortunately, we have not yet been able to determine a systematic pattern for this and must admit that this different behavior is still a mystery for us as well. For the selection of a wind generator type, we have been able to gather experience for different wind conditions. For example, very robust and slow models are not well suited in low wind regions. Wind sensitive models are often destroyed in stormy regions without appropriate modification (for example shortening of the wings, suitable brakes and switching behaviour).
* * *
L245 – do you mean discharging rather than charging?

Yes. Changed as suggested.

---

## Author Comment (AC2)

First of all, we would like to thank Fabian Walter for his time and constructive, thorough and helpful suggestions, which are each addressed below. Our responses are organized in the following color code:

- the original text of the reviewer (black)
- response to the reviewer comments (blue)
- text removed from the main article (lila)
- text added to the main article (green)
* * *
This submission by Eckstaller et al. proposes technical solutions for deploying on-ice seismic broadband stations in Antarctica. The authors present a setup that solves some of the technical problems implied by the rough Antarctic environment and offer an outlook on future developments and additions to their system to tackle other challenges like limited sunlight during the Austral winter.

The manuscript is clearly written and easy to follow. As someone who has installed seismometers in icy conditions throughout his scientific career, I welcome such a communication. Written and publically available documentation on technical details of instrument deployments can be extremely important for future projects and may make the difference between research success and failure. At the same time, I would encourage the authors to make some major modifications on how this material is presented. I detail these points of criticism below.

Fabian Walter.

MAJOR COMMENTS

An easily implemented though to me essential change would be not to mislead the reader in thinking that a technical solution to year-round broadband station deployment has been successfully tested. This is suggested in the title and the last paragraph of the introduction. In the final paragraph of the Discussion, the authors back down from this claim stating that bridging the winter gap was not yet the goal of this technical solution but is left for future efforts. This will likely annoy the reader who truly looks for useful information for his/her future deployment and feels encouraged by the abstract and introduction. At all parts of the manuscript, the reader should be clear on what this study is for. To my mind there is nothing wrong with presenting an intermediate step to an ideal broadband setup, but this has to be communicated from the beginning.

We agree with the reviewer that this is misleading. We changed the title to "**Towards a** self-sufficient mobile broadband seismological recording system for year-round operation in Antarctica"

With respect to the last paragraph of the introduction, we changed our message in a way that we: "**present strategies** to get the system through the sunless polar winter".
* * *
My second point of major criticism is that the material tends to be presented as an experience report rather than a systematic evaluation of different options, which I would expect for a scientific paper. It would help to see more numbers, especially on power consumption, that were the basis for the hardware choices. See my specific comments below.

We fully agree with the reviewer. Throughout the text, we have added information on e.g., power drain of the instruments (see new Table 1), estimates on the total power consumption over a year for a single station as well as more numbers with respect to the performance of batteries (in the text and in the new Figure 5).

Concerning our hardware choices, we would like to emphasize that we still use instruments from our older equipment pool for various reasons (e.g., acquisition costs and continuity in data). Our basic goal (as emphasized in our manuscript) is to use a seismometer station that is as energy-efficient as possible. The new components for this (e.g. the more economical Metrozet MBB2 seismometers or the Quanterra data loggers) will be taken over piece by piece into our equipment pool. For many of our permanent stations, however, we also consider it more important to keep the existing instruments to ensure the continuity of the data and save costs by continuing to use existing equipment as long as it works reliably.
* * *
Finally, I suggest providing more information on the Antarctic setting: Like I said, I have deployed many instruments on glacial ice myself, however, the majority was on ablation zones, which are ice-free in the summer. This is a very difficult environment, as well, although completely different from the Antarctic setting. For example, if the station is visited infrequently and rarely, which is the final goal of the author's set up, how do you deal with snow accumulation? Do the instruments have to be unburied and moved close to the snow surface regularly? Or is snow accumulation negligible at the locations and over the time scales of interest?

We agree and gladly provide more information on the Antarctic setting of our seismic stations. In the following we respond to the questions of the reviewer:

**How do you deal with snow accumulation?**
Snow accumulation is present at all our sites that are located on ice and at none of them we are located in an ablation regime. Accumulation on the different sites (permanent and temporary sites) ranges from 15-20 cm per year on the plateau (e.g., at Kohnen station) to almost 3 meters at VNA3 (Sörasen). This factor is the most important one that determines the frequency of station visits (e.g., once a year at Sörasen but less frequently at other sites). If the corresponding time window is not maintained, the stations have to be dug up very deep or, in the worst case, are not found anymore.

**Do the instruments have to be unburied and moved close to the snow surface regularly?**
Most of our stations (permanent and mobile stations) need to be relocated to the surface and, as stated above, the intervals depend on the local snow accumulation. However, some stations are also located on nunataks, where the equipment is not buried by snow over time and, thus, does not need to be visited to relocate the instruments. Solid fixing of the instruments and stable enclosure for the seismometer is much more important at the rock (nunatak) stations. They provide a better connection to the ground but on the downside, we have more problems with noise. The ratio between snow and nunatak stations is approximately 80/20.

**Or is snow accumulation negligible at the locations and over the time scales of interest?**
Snow accumulation is a factor that concerns all permanent stations that are not located on nunataks, which is the majority in our station network. In regard of our mobile stations, snow accumulation has

not been a crucial issue as long the deployment times do not exceed several years (depending on the accumulation at the respective site).

We added the following paragraph to Section 2 (AWI's regional seismographic network):
"The permanent and mobile temporary seismological stations of the regional AWI seismographic network are located in different glaciological regimes and thus are affected by different snow accumulation rates. None of our stations is located in an ablation area. Snow accumulation on the plateau (e.g., KOHN at Kohnen station; Figure 1b) ranges between 15-20 cm per year. By contrast, we observe several meters per year of snow accumulation at the coastal stations (e.g., 3 m per year at VNA3). Depending on the local snow accumulation, the components of the seismological stations, as well as the solar panels or masts, must be relocated to the ice surface, otherwise they will be buried by the snow over a longer period. This action is mandatory once a year for VNA3 on Sörasen and every 3-5 years for the stations on the plateau. Some stations (e.g. UPST, SVEA, WEI; Figure 1b) are located on nunataks where we observe neither significant snow accumulation nor ablation."
* * *
SPECIFIC COMMENTS

The manuscript has a few typos and grammatical mistakes (in particular the use of past tenses) that should be corrected with a thorough proofread.

"Data" is plural.

Agreed and changed where we used "data" in the singular form.
* * *
Line 37: rewrite: "However, there is little on ...".

We agree that there is a mistake in that sentence. We changed it to:
"However, there is only little seismic activity originating from on the Antarctic plate itself due to a low level of tectonic activity (Sykes 1978)."
* * *
Line 53: "adequate capacity" should be defined.

We changed it to: "effective capacity".
* * *
Line 75: "ice lying over solid rock": as opposed to what? Lying over subglacial till?

Thank you for pointing this out. We are not much more precise:
"The data quality of VNA2 and VNA3 is substantially better than VNA1 data because the latter is stationed on the ice shelf and the former two are stationed on grounded ice."
* * *
Figure 2 and associated text: The wind battery box still seems an idea rather than an established solution. This should be made clear from the beginning of the manuscript. What is the difference between an equipment and an electronic box?

The wind battery box is not just an idea, we have implemented this system at several stations (VNA2, VNA3, UPST, SVEA and KOHN). However, the reviewer has a point because we are not clear throughout the text for which stations the battery box is implemented and for which stations it is not. Typically, the wind battery box is only implemented at stations that run for a longer time and not for our short-term mobile stations. This is actually mentioned in Section 4.2.1 and is also written in Figure 2 (below the wind battery box), however, we now added this information also to the figure caption in Figure 2:

"Note that the station design for our mobile stations does not include the wind battery box. The wind battery box is, however, part of our permanent stations (VNA2, VNA3, SVEA, KOHN and UPST; Figure 1b)."

Concerning the question about the difference between the equipment and electronic box, we assume that the reviewer is referring to the legend in Figure 2. The equipment boxes are those where for example the batteries and data loggers are stored (Peli Case; e.g., those in Figure 3d,e). They have a light gray background and a blue border color. The electronic boxes are stored inside the equipment boxes and have a darker gray background color and a black border color (e.g., those in Figure 4a,b).
* * *
Table 1: This table would strongly benefit from numbers, especially on power consumption or supply (which is given in the text for some elements). Such a concise presentation would be extremely useful for a reader interested in adopting the author's approach or parts of it.

This is a great idea and we agree that this information was missing. We added the power drain for all instruments now in Table 1.
* * *
Lines 103-106: Here some numbers about the power consumption for different configurations would be more helpful than just specifying the authors' favorite choice.

Agreed and numbers added as suggested.
* * *
Figure 3A: Is the battery box associated with solar or wind power? Later from the text it is clear that it's the former, but it should also be stated in the figure or its caption.

Agreed. We now write "solar battery box (e)" in panel (a) of Figure 3.
* * *
Line 152: As a non-expert I would expect a reference for this statement.

Done. We have added as a standard reference for all our statements about batteries the technical manual for batteries of Lifeline:

https://lifelinebatteries.com/wp-content/uploads/2015/12/6-0101F-Lifeline-Technical-Manual-Final-5-06-19.pdf
* * *
Section 3.1.3: This paragraph is held rather general and superficial. Many readers would be interested at which temperature range or minimum temperature battery heating is worth it and when it costs more than it provides gains in terms of power supply. Ideally, this information should be given here and backed up with numbers, e.g., from test measurements.

We agree that this would be helpful. However, our battery heating system only use excess energy from the solar panels (therefore it doesn't "cost" anything). Until now, we have made no tests to evaluate how battery heating in winter (with energy from the batteries themselves) would increase the performance of the available capacity. This type of system would be desirable but is not implemented for our battery heating. From our point of view this is clearly stated in the text:
"Therefore we realized the option for battery heating if sufficient power is available."
Hence, unfortunately, we are not able to provide the information requested by the reviewer.

Nonetheless, we added a few more sentences to this section to make clear what we mean. We also added an extra figure (Figure 5) where we display a curve on battery capacity vs. temperature and electrolyte freezing point vs. temperature. The data from these plots were taken from the Lifeline Technical Manual for Batteries.
* * *
Line 157: Not sure what is meant by "which can switch 6 amperes of heating current".

We agree with the reviewer that this statement is confusing.

It means that the switch permits a heating current of up to 6A. This implies that we have to make sure with the heating plates that not more than 6A can flow, which corresponds to a maximum heating power of 72 W. Our heating plate set includes 2 times 15 W, which is well below that.

This information has now been added to the corresponding part in the text.
* * *
Line 167: What is meant by "switched opposite to each other"?

We mean that they function in a way that power only comes from the battery box with the higher voltage.

We changed the sentence accordingly: "For two connected solar battery boxes the BBat-controller acts as two ideal Schottky diodes which function in a way that power comes only from the battery box with the higher voltage."
* * *
Lines 174-175: What is meant by "cascading batteries"?

We mean that the amount of total battery capacity in the system can be extended through additional battery boxes. We modified the text accordingly.
* * *
Lines 184-185: This sentence reads more like an instruction manual than a scientific piece of text.

Agreed. We changed the sentence accordingly:

"If using a Quanterra Q330, the cable from the rear sensor connectors needs to be configured appropriately for this recorder."
* * *
Lines 194-195: Is this based on test measurements? Or is there a reference for this statement?

Thank you for pointing this out. We agree that the sentence is not fully clear (and the initial statement was not correct) and we added more information from the technical manual for Lifeline batteries. We added a reference with respect to the usable effective capacity of our AGM batteries (new Figure 5) when fully charged at 25°C, which is ~ 30% at -40°C. The statement that the remaining power after polar winter onset (if solar panels are the only power source) lasts for approximately 14 days for one battery is based on our experience. Therefore, we now state:

"Based on our experience, when considering a usable effective capacity of 30% at -40°C (Figure 5), one 125 Ah AGM battery can provide power for approximately 14 days at polar winter onset (if solar panels are the only power source)."

**Source:**
Lifeline Technical Manual: For Lifeline Batteries (2019). Concorde Battery Corporation, https://lifelinebatteries.com/wp-content/uploads/2015/12/6-0101F-Lifeline-Technical-Manual-Final-5-06-19.pdf
* * *
Line 196: delete "additionally"

Done.
* * *
Line 201: use of "huge" is awkward.

Agreed. We changed "huge" to "large".
* * *
Lines 210-211: Why can Li batteries not be recharged? Because of low temperatures?

True, in principle, it is possible to use rechargeable lithium batteries but we are also aware that charging lithium batteries at very low temperatures can be complicated. In addition, next to the increased costs, this would make the overall system more complex and larger, which we want to avoid.

Regarding our Li-battery concept, we are talking about additional (fully charged) batteries installed during assembly or maintenance only for the polar night. Whether these are chargeable or not makes no difference and is only a question of cost and sustainability. They are replaced with full batteries at the next maintenance (or, in the case of mobile stations, the station is removed and reassembled at another location after one or two years). If they were left to charge in the field throughout the year, additional charging resources would be needed. Then they would resemble the same system that is already existing: solar cells and rechargeable batteries.
* * *
Lines 217: Which remaining energy?

We are referring to the remaining energy (or better excess energy) from the wind generator.
We adapted the sentence accordingly.
* * *
Line 218: "easily integrated" sounds a bit fuzzy. The reader is left wondering why then it hasn't been integrated (see following sentences).

Agreed, we removed "easily".
* * *
Lines 224-225: Why is the wind turbine installation so involved?

We highlight the wind generator installation procedure because with respect to the mobile station it consumes a considerable amount of time, which is proportionally large considering the fast deployment of the solar panels, seismometer and connecting the boxes. We added this now to the corresponding sentence.
* * *
Line 232: The use of adjectives like "careful" and "proper" weakens this part of the paper. More specifics would help.

We agree and removed "careful" and "proper" from the sentence. However, we believe that more specifics on the solar controller that we decided not to use anymore, would be also confusing for the reader. Hence, the sentence reads now more neutral:

"The SB 3000i offers a variety of features and needs comprehensive programming for the setup."
* * *
Line 241: Quantify "very low temperatures".

Agreed. We added: "-20 to -40°C".
* * *
Line 244: Is "mass" the right word here? Better "ground"?

Agreed. Changed: "mass" to "ground".
* * *
Section 4.3: A general remark: we also had big problems with static charge on Alpine glaciers. In the end we found it beneficial to make sure all station elements (sensor, digitizer, metal frames, ...) were connected and at the same potential. This solved most problems although no ground was available on the ice.

Thank you for the remark. With regard to connecting all instruments at the same potential (minus), we had initially the problem that this was not possible with the BlueSky solar controller because it messed up the charge controller. This problem does not exist with the Morningstar controller and we have been doing it this way since. We also experienced that the system failure after connecting all elements to the same potential was less frequent, however, the discharge problem remained in general.

Although this is just a general remark from the reviewer, we added the information stated above to the text because we find it to be useful for the reader.

"Additionally, by connecting all instruments and equipment to the same potential (minus), the damage or failure rate of the system due to electrical discharge was reduced. It should be noted that this is not possible with all solar charge controllers (for example, it was not possible with the Blue Sky solar charge controller but possible with the Morningstar solar charge controller)."
* * *
Line 253: rewrite "partly very different"

We changed the sentence accordingly:
"The component design and the available resources of various temporary or long-term year-round seismic measurements in Greenland (e.g., Dahl-Jensen et al., 2010) and Antarctica (e.g., Hansen et al., 2015) differ between surveys."
* * *
Line 255: "fast to deploy" with respect to what?

With respect to using the limited time in the field efficiently. We changed the sentence accordingly:

"Our concept of a fast to deploy, compact, modular self-sufficient mobile seismic station aims to use the limited time in the field efficiently and is based on many of the experiences described in the literature, which we discuss in the following."

Lines 274-278: Here the original goals are redefined. This has to be changed.

We agree and removed the paragraph due to its redundancy.
* * *
Line 291: In this sentence the term "polar" or "on-ice" or something equivalent should appear.

Done. We added "polar regions".

Lines 294-295: This statement is trivial.

We agree and removed this statement.

Lines 296-298: Perhaps it's worth considering that seismic records have high sampling rates and thus the setup could easily afford power supply for GPS, temperature gauges, and other environmental monitoring logging at lower sampling rates.

We appreciate the idea of the reviewer and modified the sentence accordingly:

"Our system concept is not specifically limited to the application to seismology stations (except for noise suppression) and can also be extended by additional instruments with low power consumption (e.g., to monitor environmental parameters)."
* * *
General question: Would it be possible to stream data with this setup or is this more than an incremental step in power consumption?

Yes. At VNA2 and VNA3 data is streamed in real-time over terrestrial data radio to Neumayer Station. However, this is only possible because these two stations are located relatively close to Neumayer Station (< 100 km; Figure 1b) and the transmission path is free of obstacles. In principle, the data transmission over iridium, other satellite systems, or via repeater stations would be possible for our other stations. However, this requires new equipment, additional power, is more expensive, and is thus not planned at the moment.

As this comment is labeled as a "general question" we performed no modifications in the text.
* * *
There are two sections 3.1.1. In general, I find this manucript contains many sub and sub-sub sections given its limited length.

We agree and changed the section number accordingly. The comment about too many subsections in the text is justified. However, one can also argue that it is a matter of taste to list more or fewer subsections. If this is not a major issue for the reviewer, we would like to keep the number of subsections. We argue that this provides a better structure in the manuscript and also makes it easier for the reader to find his way through the text.

---

## Author Comment (AC3)

First of all, we would like to thank the reviewer for his time and constructive, thorough and helpful suggestions, which are each addressed below. Our responses are organized in the following color code:

- the original text of the reviewer (black)
- response to the reviewer comments (blue)
- text removed from the main article (lila)
- text added to the main article (green)
* * *
This work highlights an effort to develop robust seismic stations for Antarctica. As noted here and in previous work, operating seismic stations (or any other autonomous geophysical station) is challenging due to the environmental conditions, in particular the cold and dark.  While all sharing of technological advancements are welcome, I feel that in this case, the authors have provided too few details to make the manuscript useful to the audience. I would encourage the authors to provide more information so that other researcher can more easily replicate their design. Below I have a few detailed comments.

We would like to thank the reviewer for the generally positive assessment of our manuscript and the helpful suggestions. We agree with the reviewer that more details are needed and will make our manuscript much more helpful to other researchers. Our responses to the detailed comments raised by the reviewer can be found below.
* * *
Detailed Comments:

Line 58: Perhaps a bit picky, while -20 C to -40 C is certainly cold, I'm not sure it is "extreme" since a Nanometrics trillium posthole 120 can operate to -50 C.

Thank you for this hint, which is not picky at all. Our temperature range from -20 to -40°C represents the average temperature range for operation. The temperatures can be warmer in summer and slightly colder in winter. We added this now in the main text. However, because most of our stations are buried in the snow, the temperature fluctuations are less extreme.
* * *
Line 75: Mention that Neumeyer Station is on an ice shelf ( I had to look this up)

We fully agree. The information that VNA1 is positioned on an ice shelf is missing. We have added this now in our new version.
* * *
Line 93: Is it better to use the term Peli (or Pelican) case? A websearch for "Eurocase" doesn't led to the product, I only got to the product page by search the model number included in table 1.

We agree and want to thank the reviewer for his thorough research. Our boxes were originally ordered in a catalog and were termed "Eurocases" by the supplier. But the reviewer is completely right, that the correct term is "Peli ISP2 CASES - Inter-Stacking Pattern Cases". We changed this accordingly in the text and in Table 1.
* * *
Line 93: I notice on the spec sheet for EU080060-5010 that a minimum temperature is -30 C, have the authors had experiences with this product at colder temperatures?

The boxes themselves have experienced lower temperatures. However, as most operations where we handle and move the boxes occur mostly during the summer season when it is much warmer. Therefore we have only a little experience in terms of how they would behave under stress at temperatures below -30°C. Our station at Kohnen is exposed to an almost constant temperature of -40°C and less and we didn't notice any brittle damage to the boxes so far.

Hence, we assume that the boxes can be used at temperatures below -30°C. We speculate that the company has only a very small number of customers that use the boxes for these conditions. Therefore, tests on the material are probably not worthwhile for temperatures below -30°C degrees.
* * *
Line 110: More details on the XEOS XI-202 on the SeiDL Controller are needed? What exactly operations can they perform do? How much power do they consume?

We added the power drain values for the Xeos and SeiDL controllers to Table 1. The operations both controllers perform are (i) to read SOH (state of health) data from the recorders and (ii) send SOH data as short burst data (SBD) over iridium as an email once per day. This is already stated in the text (Section 3).
* * *
Line 116/(table 1). What are the characteristics of the GPL31XT batteries that led the authors to choose them? What differentness them from other AGM batteries?

Our GPL31XT batteries have the standard advantages of AGM batteries. In the past, we used GPL31M batteries with 105 Ah. The GPL31XT batteries have the advantage that they provide 125 Ah at the same size and weight (~ 30 kg) as the GPL31M type. We've also chosen this battery type because it is still possible to be carried and handled by one person. We added this information now in Table 1 in the "Comment" column. Please note that the type names "GPL31XT" and "GPL31M " are the product names from the company "Lifeline".
* * *
Line 123: Quantify high wind. I have seen many "mechanically robust" pieces of equipment blown apart by wind.

We define "high winds" as wind speeds beyond the range where it is feasible to work outside (25-50 m/s). We added this number now in the text.

We have customized our wind generators to a large extent to deal with the strong winds in order to minimize the stress on the material: e.g. oversized E-brakes, shortened rotor blades and longer pauses between braking and restarting during storms.
* * *
Line 124: I think the authors mean "Pladur Panel Alveo" when they say "Alveo". When I do a web search for "Alveo" this is the only company that appears to make panels. More details one the exact nature of the panels would be nice as well.

Yes, we agree that more detailed information is needed here. The material that we use is called "Alveobloc" and is produced by Sekisui Alveo. We use three different thicknesses 3, 5 and 10 cm to create insulation blocks of 8 and 10 cm, respectively. The Alveobloc material is available in different densities and we use a softer Type 3600 (28 kg/m$^3$) and a harder type 1700 (60 kg/m3). This additional information has now been added at the corresponding place in the text.
* * *
Section 3.1.1 and 3.2- I think more details are needed (wiring diagrams?) for these sections to be helpful to the reader or an engineer.

This is a very good suggestion. We now complement Figure 4 with two wiring diagrams for the solar electronic box and recorder electronic box.

[Figure]
* * *
Section 4.1.1: A reader unfamiliar with Antarctic seismology may think this has not been successfully implemented when in fact the use of Li batteries have been the power stations for numerous experiments support the PASSCAL instrument facility (this is briefly mentioned in section 4.4). Thus, I think a reference to Hansen et al., 2015 (where the use of Li Batteries is explicitly stated) is needed in this section.

We agree. We added the following sentence in Section 4.1.1:

"The concept of using Li-based batteries has already been successfully demonstrated in Antarctic campaigns with the PASSCAL instruments (e.g., Hansen et al., 2015)"
* * *
Section 4.1.2: More details are needed. I have talked to many people whom have had NO success with wind generators in Antarctica for various reasons (wind extremes, icing). If the authors are utilizing wind power successfully at VNA2 and VNA3 that is a great advance and I would like to know more! What are the temperature conditions? How much power is produced?

Many thanks for the interest. Of course, we have to admit that we have also experienced failures in terms of wind generator usage in Antarctica. However, our wind generators on VNA2 and VNA3 run mostly reliably and produce enough power (in fact, way too much) to ensure data recording through the winter most of the time.

We use helical horizontal axis wind turbines (HAWT). This wind generator type has already proven itself on the predecessor station of Neumayer III (Neumayer II), but in a larger version. We use the smallest version, which generates 300 Watt. The system is characterized by a very robust and simple generator and consists of three rotor blades. We have additionally shortened these to half their length to reduce the forces acting on the material (provides around 150 Watt). An advantage of these generators is that the bearings do not require regular oiling. On VNA3, the generator has run for 5 years without maintenance. However, we still see the potential for development in the control of the generator (for example, regulation in very strong winds). In addition, for our mobile stations, this type of generator is still oversized and very heavy. Here, a smaller wind generator with a horizontal rotor shaft would be more suitable.

The temperature at both VNA2 and VNA3 ranges from 0°C in summer to -50°C in winter with wind speeds up to 50 m/s.

The additional information about our wind generator types and experiences were now added to Section 4.1.2.